# Serum Factors in Primary Podocytopathies

**DOI:** 10.3390/antib14040082

**Published:** 2025-09-28

**Authors:** Edward John Filippone, John L. Farber

**Affiliations:** 1Division of Nephrology, Department of Medicine, Sidney Kimmel Medical College, Thomas Jefferson University, 2228 south Broad St, Philadelphia, PA 19145, USA; 2Department of Pathology, Sidney Kimmel Medical College, Thomas Jefferson University, Philadelphia, PA 19145, USA; john.farber@jefferson.edu

**Keywords:** podocytopathy, minimal change disease, focal segmental glomerulosclerosis, suPAR, cardiotrophin-like cytokine 1, angiopoietin-like 4, anti-nephrin antibodies, APOL1

## Abstract

Primary podocytopathies, including minimal change disease (MCD) and focal segmental glomerulosclerosis (FSGS), are caused by a circulating factor or factors injurious to the podocyte. An immunologic origin seems likely based on responsiveness to corticosteroids or other immunosuppressive agents, including calcineurin inhibitors targeting T-cells and rituximab targeting B-cells. Potential non-antibody-mediated circulating factors have been identified, including cardiotrophin-like cytokine 1, soluble urokinase plasminogen activator receptor, and angiopoietin-like 4, among others. More recent research supports a primary antibody pathogenesis, with anti-nephrin antibodies found in a significant percentage of cases. Such antibodies also predict recurrence after transplantation. Other potential antigenic targets besides nephrin include annexin, the proteosome, podocin, and CD40. Additionally, high-resolution confocal microscopy has identified punctate immunoglobulin deposits along the slit diaphragm and podocyte cell body that may or may not colocalize with abnormal punctate nephrin staining and may correlate with detectable circulating antibodies. The success of rituximab in observational studies in both native kidneys and transplants supports a primary role for autoantibodies. We discuss in detail the data supporting putative non-antibody circulating factors, as well as the recent data supporting antibody pathogenesis, which may provide some clues on treating the individual patient.

## 1. Introduction

Idiopathic nephrotic syndrome (INS) refers to the sudden onset of complete nephrotic syndrome (nephrotic-range proteinuria with hypoalbuminemia and edema) in the absence of a secondary cause. Most cases of INS are caused by primary disease of the podocyte, currently classified as either minimal change disease (MCD) or primary focal segmental glomerulosclerosis (FSGS). Both MCD and FSGS are characterized ultrastructurally by extensive podocyte damage with disruption of the actin cytoskeleton, diffuse foot-process effacement (FPE), vacuolization of the podocyte body, and villous transformation of the cytoplasm. More severe injury may result in podocyte detachment, cell death, and podocyte depletion. FSGS additionally has one or more segments of sclerosis/hyalinosis and has been subdivided into five subtypes by light microscopy, including collapsing glomerulopathy, the tip lesion, the cellular lesion, perihilar FSGS, and FSGS not otherwise specified [1]. Primary FSGS is differentiated from secondary FSGS by the presence of full-blown nephrotic syndrome (hypoalbuminemia), diffuse foot process effacement (>80%), and lack of a known secondary cause. Herein, we consider only primary FSGS.

MCD is overwhelmingly (>85%) sensitive to steroids, but it has a high chance of recurrence, including frequent relapses or steroid dependence [2]. FSGS responds to steroids in 50–60% of cases and may also relapse [3,4]. Steroid-resistant cases may respond to second-line therapy, but, if resistant, they may progress to ESKD, with recurrence after transplantation in about 30% of cases [5]. Controversy remains as to whether MCD and FSGS are distinct diseases or represent the spectrum of a single disease, with a more severe insult having segmental sclerosis, steroid resistance, and progression to ESKD [6].

The pathophysiology of these two podocytopathies is incompletely understood. We discuss them together given their shared features, with separation where possible. In 1972, Shalhoub proposed that MCD, at the time called lipoid nephrosis (L.N.), was the result of a circulating factor elaborated by T-cells, although in his conclusions, he stated that “patients with L.N. need not comprise a homogeneous group, since several causes could produce the same end-result” [7]. The search for the putative pathogenic factor(s) has since continued for MCD and evolved to include cases with detectable sclerosis, i.e., FSGS. Given the absence of ultrastructural immune deposits in both MCD and FSGS, earlier research centered on potentially toxic circulating factors. New research has brought to light potential pathogenic autoantibodies. Herein, we review prior research on circulating non-antibody factors (Table 1), as well as new data on antibody pathogenesis (Table 2 and Table 3).

## 2. Non-Antibody Circulating Factors

### 2.1. Vascular Permeability Factor

Intradermal injection of supernatants of concanavalin-A-stimulated lymphocyte cultures from nephrotic patients into guinea pigs increased vascular permeability [21]. The active factor had a molecular weight of 12kD [22]. Tomizawa et al. found that T lymphocytes from MCD patients in culture spontaneously produced a vascular permeability factor even without concanavalin-A stimulation [23]. Others confirmed a T-cell origin for such a factor [24]. Bakker and van Luijk reviewed earlier work on these and other potential factors and commented on the conundrum of whether any is/are the actual causes or are merely the result of nephrosis [25].

### 2.2. Hemopexin

Cheung et al. found a vasoactive plasma factor in active MCD with a molecular weight of 100 kD, termed 100KF, that was capable of inducing proteinuria in rat kidneys [26]. This factor was identical to hemopexin, an acute-phase reactant produced by the liver with serine protease activity [27]. Exposure of podocytes in vitro to hemopexin caused reversible actin cytoskeleton reorganization, an effect that required nephrin expression, as well as increasing albumin permeability across glomerular endothelial cells and reducing glycocalyx [28].

### 2.3. The FSGS Factor

Savin et al. developed an in vitro method to determine the so-called glomerular albumin reflection coefficient (σ_albumin_) using volume changes in isolated rat glomeruli in response to an oncotic pressure gradient by decreasing the bathing solution albumin concentration from 5 g/dL to 1 g/dL. As fluid moved into the glomeruli, the volume change could be measured. If no increased permeability occurred, the ratio of volume change of experimental serum/control serum would be 1. Abnormal permeability would result in no volume change, as albumin would cross the membrane to dissipate the oncotic gradient. The ratio of volume change of the experimental plasma to control would be minimal, approaching 0. The albumin permeability (P_alb_), a dimensionless variable, was defined as 1 − σ_albumin_ (values ranged between 0 and 1, with normal being 0 and 0.5 considered a reasonable cutoff for increased permeability) [29]. Using this assay, they compared 33 kidney transplant recipients with recurrent primary FSGS to 23 recipients without recurrence, as well as 44 non-transplanted patients with FSGS, 9 with steroid-sensitive nephrotic syndrome (SSNS), 5 with membranous nephropathy (MN), and 9 controls [30]. The recurrent group had higher P_alb_ (0.47 +/− 0.06) versus the controls (0.06 +/− 0.07) and patients with SSNS (−0.15 +/− 0.05), MN, or non-recurrent FSGS (0.14 +/− 0.06); reductions occurred following plasmapheresis in the recurrent group (0.79 +/− 0.06 to 0.10 +/− 0.05). A P_alb_ value of > 0.50 provided a reasonable separation of recurrence versus non-recurrence (86% versus 17%).

By contrast, studying patients with steroid-resistant FSGS in native kidneys receiving plasmapheresis, Feld et al. found no relation between P_alb_ reduction following pheresis and a change in proteinuria [31]. Cattran et al. followed the serial P_alb_ levels in patients with steroid-resistant FSGS randomized to cyclosporin or placebo and found no relation between P_alb_ and remission or relapse of proteinuria, and the antiproteinuric effect of cyclosporin was not related to changes in P_alb_ [32].

The FSGS “factor” responsible for changing P_alb_ was explored by Sharma et al., who utilized discarded plasmapheresis fluid from patients with recurrent or refractory FSGS. The “factor” was found to be a non-immunoglobulin protein with a molecular weight of 30–50 kD [33]. It is anionic, binds to protein A, and reduces tyrosine phosphorylation [34]. Its effect can be blocked by cyclosporin [35], normal serum [36], apolipoproteins [37], indomethacin [38], and *Tripterygium wilfordi* multiglycoside [39]. Subsequent work by this group identified cardioptrophin-like cytokine factor-1 (CLCF1), a member of the interleukin-6 family, as a potential candidate. CLCF1 produced progressive increases in P_alb_ similar to exposure to serum from patients with recurrent FSGS, which was blocked in both instances by anti-CLCF1 antibody [40]. Interleukin-6 itself did not increase P_alb_.

### 2.4. Zinc Fingers and Homobox Transcription Factors and Angiopoietin-like 4

The zinc fingers and homobox (ZHX) family of transcription factors (ZHX1,2,3), in association with angiopoietin-like 4 (Angl4), have been studied as mediators of MCD/FSGS [41,42]. Normally forming cell-membrane-associated heterodimers involving ZHX2 (ZHX1-ZHX2 on the podocyte body tethered to the cytoplasmic tail of aminopeptidase A and ZHX2-ZHX3 at the slit diaphragm tethered to ephrin B1), abnormal ZHX nuclear expression may underly podocytopathies. Increased ZHX1 in MCD and increased ZHX3 (+/−ZHX2) in FSGS have been found [41].

Low podocyte ZHX2 expression has been shown in patients with both MCD and FSGS [41]. In the ZHX2 hypomorphic state, there is both enhanced HBX1 nuclear expression and increased membrane binding of ZHX1 to the IL4-receptor. The former may enhance susceptibility to the mild cytokine storm of the common cold, a known trigger of MCD relapses, and the latter provides a linkage between chronic atopy (high soluble IL4 receptor) and MCD [42]. Regarding genetic susceptibility, an insertion/deletion upstream to the human HBX2 gene induced a reduction in ZHX2 in a cultured human podocyte cell line and was found in some patients with either MCD or FSGS [43].

In MCD, angiopoietin-like 4 (Angl4) production is increased, and in FSGS, Wilm’s tumor-1 (WT1, a crucial transcription factor) and other factors are decreased [41,42]. Angl4, an inhibitor of lipoprotein lipase secreted by adipose tissue, skeletal muscle, the liver, and the heart, circulates at increased levels in all forms of nephrosis, promoting hypertriglyceridemia. Angl4 can reduce proteinuria in experimental animals in a negative feedback loop [44]. Podocytes constitutively express Angl4 at low levels, which are significantly increased in MCD, where it is produced in a hyposialylated form which significantly contributes to podocyte injury and proteinuria [42,45]. Angl4 binds to β_1_ and β_5_ integrins on podocytes (α_3_β_1_) and glomerular endothelial cells (α_V_β_5_) with increased affinity when hyposialylated [42]. It does not bind β_3_ integrins [46]. Binding may also occur through integrin-independent mechanisms (syndecans and proteoglycans). It remains uncertain whether damage is due to integrin or non-integrin binding [42].

In mice, the angl4 gene is glucocorticoid-sensitive, supporting its role as a mediator of MCD [45]. In an Angl4 knockout mouse model of lipopolysaccharide-induced nephrosis, the downregulation of podocin and alpha-actinin-4 was mitigated compared to wild-type mice and proteinuria was significantly less [47]. In one study, neither serum nor urine Angl4 levels could differentiate patients with MCD versus FSGS or MN [48], presumably because circulating normally sialylated Angpl4 is produced extra-renally compared to pathogenic hyposialylated angl4, which is predominantly produced locally by the podocyte [42]. Normally sialylated Angl4 reduces proteinuria by binding to endothelial α_V_β_5_ integrin [44]. However, in one study, restricting measurements to circulating Angl4 with high pI (hyposialylated), only patients with MCD in relapse had elevated levels compared to patients with MCD in remission, FSGS, or MN, data that require confirmation in larger studies [45].

### 2.5. suPAR

The urokinase-type plasminogen activator receptor (uPAR), tethered to the plasma membrane via glycosylphosphatidylinositol (GPI) and produced by the *PLAUR* gene, is normally expressed on immune cells and endothelial cells. Upon binding urokinase, plasminogen is cleaved to form plasmin extracellularly [49], with a separate binding site for vitronectin. uPAR lacks transmembrane and cytoplasmic domains, and intracellular signals are transduced by its interaction with other receptors, including β_1_ and β_3_ integrins [49]. Such signaling is involved in immune activation, remodeling, and migration [50]. uPAR contains three domains, DI, DII, and DIII [49]. Cleavage of uPAR from GPI results in the soluble form suPAR that contains all three domains (DI, DII, and DIII), which can be further cleaved by various proteolytic enzymes to result in at least two additional circulating forms (suPAR2-3 containing DII and DIII and lacking DI, and suPAR1 containing only DI) [50]. The role and pathogenic significance of these fragments needs clarification [51].

Elevated levels of suPAR are found in states of inflammation and immune activation, including sepsis/SIRS, tuberculosis, malaria, HIV, COVID-19, autoimmune diseases, and cancer. Serving as a nonspecific marker of inflammation, elevated levels of suPAR have prognostic significance [50]. Significantly elevated levels are found in chronic hemodialysis patients [52]. Hayek et al. studied a large cohort of adults with cardiovascular disease (median suPAR level of 3040 pg/mL) and found an independent association with incident chronic kidney disease and a faster rate of decline in eGFR, particularly in patients with a normal baseline eGFR [53]. Elevated suPAR levels correlated with increasing albuminuria and other complications in patients with type 1 diabetes [54].

Podocytes express uPAR, which interacts with β_3_ integrins, and suPAR can activate these integrins (specifically α_V_β_3_) to affect adhesion and migration [55]. Wei et al. studied the suPAR levels in patients with FSGS compared to other glomerulopathies and found that suPAR was elevated above 3000 pg/mL in two-thirds of patients with primary FSGS. They also showed in mouse models that suPAR activated podocyte β_3_ integrin to cause FPE and proteinuria [56]. suPAR concentrations were significantly elevated only in FSGS and not in MCD (in relapse or remission), MN, preeclampsia, and controls. Furthermore, the highest levels were found in patients destined for recurrence after transplantation, as compared to those without recurrence. In two additional cohorts of patients with FSGS, 84.3% and 55.3% had levels above 3000 pg/mL compared to 6% of controls [57]. Hence, suPAR was promulgated as a circulating factor responsible for primary FSGS, especially cases destined for recurrence after transplantation. The primary source for increased circulating suPAR was found to be bone-marrow-derived immature myeloid cells (not the lymphocyte lineage originally proposed by Shalhoub [7]) based on various models in experimental animals [58], suggesting that cases of primary FSGS induced by suPAR represent a systemic disease [58].

Studying human podocytes in vitro, Alfano et al. showed that full-length suPAR and not suPAR2-3 (DII, DIII fragment) activated α_V_β_3_-integrin to down-modulate nephrin expression via repression of the critical podocyte transcription factor WT-1. An α_V_β_3_-inegrin blocker abrogated this effect [59]. Infusion of suPAR into *Plaur*^−^ mice reduced WT-1 and nephrin expression and induced proteinuria.

Contradictory results have been found by others. Cathelin et al. injected two forms of mouse suPAR into wild-type mice and indeed found glomerular localization, but no significant proteinuria or FPE [60]. Similarly, Spinale et al. found that neither injection of recombinant suPAR in the short term nor transgenic suPAR overexpression in the longer term was able to induce significant proteinuria in mice [61].

The specificity of elevated suPAR levels for FSGS has been questioned. Multiple studies have found that serum levels of suPAR were unable to differentiate primary FSGS from other glomerulopathies, such as secondary FSGS, MCD, and MN [61,62,63,64,65]. In most studies, suPAR levels correlated directly with inflammatory markers (CRP), thereby confirming the relation to inflammation, and were inversely correlated with eGFR in essentially all studies, indicating reduced clearance as a significant factor for elevated levels.

For example, in a cross-sectional North American pediatric study, Bock et al. found no difference in suPAR levels between patients with FSGS, other glomerular diseases, and healthy children [62]. Likewise, in a study of children from India, Sinha et al. found no difference in suPAR levels between patients with steroid-resistant FSGS, steroid-resistant MCD, and SSNS [65]. In a study of Chinese adults, suPAR levels could not differentiate primary from secondary FSGS [63]. In a study of Dutch adults, Meijers et al. stratified patients by eGFR, the strongest determinant of suPAR levels, and could not differentiate active FSGS, FSGS in remission, and non-FSGS CKD controls [64]. In the North American NEPTUNE cohort, neither serum nor urine suPAR could differentiate FSGS versus MCD, MN, and IgAN [61]. Patients with genetic FSGS also had elevated levels, higher than those with presumed primary FSGS [56]. Yoo et al. found elevated levels of suPAR in patents and experimental animals with diabetic nephropathy [66]. Interestingly, podocytes from patients with diabetic nephropathy overexpressed sphingomyelinase-like phosphodiesterase 3b, which, in vitro, interfered with suPAR activation of β_3_ integrin, thereby fostering an apoptotic podocyte phenotype [66]. As an aside, sphingomyelinase-like phosphodiesterase 3b may bind and be stabilized by rituximab to mediate its beneficial effects independent of B-cell depletion [67]. In IgAN patients, higher suPAR levels significantly correlated with S (segmental sclerosis) lesions [68].

Overall, these conflicting data suggest that suPAR is most likely involved, along with other factors (*vide infra*), in the pathogenesis of podocyte injury in primary FSGS. Elevated levels are not specific and are not able to differentiate primary FSGS from other glomerulopathies.

### 2.6. Calcium/Calmodulin-Dependent Serine Protease (CASK)

Beaudreuil et al. analyzed eluates from protein column immunoadsorption from eight patients with recurrent FSGS and identified CASK, which was not found in the serum of healthy donors or transplant patients with non-recurrent FSGS, non-FSGS transplants, or non-transplanted but nephrotic patients with MCD, diabetic nephropathy, or MN. Recombinant CASK induced actin cytoskeleton reorganization in immortalized podocytes, with redistribution of ZO-1 from the intercellular membrane surface to form punctate intracellular deposits. Extracellular CASK was shown to bind to CD98, forming a complex capable of binding β_1_ and β_3_ integrins. Injection of CASK into mice produced proteinuria and foot process abnormalities [69].

### 2.7. Micro-RNAs

Micro-RNAs (miRNA) are small (18–22 nucleotides), single-stranded, non-coding RNAs that regulate the translation of target mRNAs and, hence, the expression of multiple genes. Additional types include circular RNAs and long non-coding RNAs (lncRNA, ≥200 nucleotides). Multiple miRNAs can target a single gene, and a given miRNA can target multiple genes. There are over 1900 miRNAs expressed in humans on the microRNA database (https://www.mirbase.org, accessed on 1 July 2025); about 700 are expressed in the kidney [70]. They can be measured in blood or urine.

Multiple miRNAs have been found to be dysregulated in primary podocytopathies. For example, high miR-196a, high miR-30a-5p, and high miR-490 in urine predicted active FSGS versus FSGS in remission. miRNA levels decreased with treatment in steroid-sensitive cases, but not resistant ones [71]. miR-193a is elevated in experimental FSGS and downregulates WT-1 and its target genes, including nephrin and podocalyxin [72]. Wang et al. found elevated urinary miR-193a in active pediatric FSGS, allowing for differentiation from active MCD and IgAN [73]. Another study found higher plasma levels of miR-193a, miR-125b, and miR-186 than in controls, but only the latter two could differentiate active FSGS from FSGS in remission [74]. Many other studies are available, and the reader is referred to recent reviews for a more in-depth discussion [70,75,76]. miRNAs may become useful as biomarkers for diagnosis and/or for response to therapy, but more work is necessary.

### 2.8. Soluble CD40 Ligand

Expressed on professional antigen-presenting cells, CD40 is also expressed on podocytes [77]. CD40 ligand (CD40L) is membrane-bound on T-cells and platelets, but it is also present as a soluble form (sCD40L), which can upregulate podocyte CD40 expression [78]. The main source of sCD40L is activated platelets [78], but activated T-cells release it as well [79]. Activation of CD40 by sCD40L may enhance podocyte production of matrix metalloproteinases on cultured podocytes [78]. Kuo et al. showed enhanced podocyte inflammatory mediator production with exposure to CD40L, an effect magnified by concurrent IL-17 [80].

Doublier et al. showed that exposure of cultured podocytes to sCD40L rapidly downregulated nephrin expression, an effect that could be inhibited by blocking CD40–CD40L interactions [77]. The actin cytoskeleton was also disrupted, and sCD40L enhanced glomerular permeability in isolated rat glomeruli. When injected into mice, sCD40L resulted in a marked decrease in both nephrin and podocin at 24 h, although there was not a significant increase in proteinuria.

## 3. Autoantibody Pathogenesis

### 3.1. Ubiquitin Carboxyterminal Hydrolase L1 Autoantibodies

Autoantibodies against several target antigens have been identified as potential mediators of primary podocytopathies. Involved in proteosomal function, ubiquitin carboxyterminal hydrolase L1 (UCHL1) is normally expressed in parietal epithelial cells of Bowman’s capsule and renal tubular epithelial cells. Podocytes had no or minimal staining in normal human kidneys [81,82,83]. Liu et al. found significantly increased podocyte staining in immune complex-mediated glomerulopathies (lupus, MN, and IgAN), but not in MCD or FSGS, both of which had minimal staining like normal tissue [83].

By contrast, Jamin et al. detected anti-UCHL1 antibodies significantly more frequently in children with idiopathic SSNS than in similar patients in remission, healthy children, children with IgA vasculitis, and healthy adults [84]. Interestingly, adults with MCD did not have elevated levels. Purified anti-UCHL1 from patients injected into mice produced proteinuria and foot process effacement. No immune deposits were noted on EM as in primary podocytopathies [84].

Chebotareva et al. compared 35 adults with apparent primary FSGS and 30 adults with MCD to adults with MN (*n* = 21), membranoproliferative glomerulonephritis (13), IgAN (22), and healthy controls [85]. Baseline anti-UCHL1 antibody levels were significantly increased with MCD as compared to the other groups. A relatively widespread level was seen in FSGS patients. Steroid-responsive FSGS cases had higher levels, and steroid-resistant cases had lower levels.

### 3.2. Proteosome Subunit Alpha Type 1 Autoantibodies

Other work supports the proteasome as the target of autoantibodies underlying INS. Ye et al. studied 341 Chinese children with INS for circulating antibodies against mouse podocyte proteins and found that 66% had antibodies against at least one of seven proteins, including proteosome subunit alpha type 1 (PSMA1) [86]. Subsequently studying 54 Chinese children with INS, they found significantly higher serum anti-PSMA1 antibodies compared to control children and patients with Kawasaki disease, IgAN, or Henoch–Schoenlein purpura [87]. Using a cutoff of >28.2, the area-under-the-curve (AUC) for diagnosing INS was 0.904, the antibody level was directly correlated with the degree of proteinuria, and the level significantly decreased in those attaining remission. Under confocal microscopy of kidney biopsies in anti-PSMA1-positive patients, punctate IgG was detected that colocalized with PSMA1. Furthermore, PSMA1 knockdown in mouse podocytes produced abnormal adhesion and migration with a disordered cytoskeleton, and deletion in zebrafish resulted in edema and foot process effacement [87].

### 3.3. Annexin A_2_ Autoantibodies

Ye et al. studied 20 Chinese children with INS and found that 14 had serum IgG antibodies against annexin A_2_, a protein directly involved in the transduction of signals from the podocyte surface to the actin cytoskeleton through Rho GTPases. This signaling allows for normal podocyte adhesion, migration, and phagocytosis [88]. Of 596 children with INS, 106 (17.8%) had such antibodies compared to 0 control children and patients with lupus or IgA vasculitis. All 61 biopsied patients had MCD/FSGS, and IgG colocalized with annexin A_2_ and nephrin by confocal microscopy. Injected into mice, these antibodies induced proteinuria and foot process effacement with downregulation of nephrin and WT1. Added to podocytes in culture, annexin A_2_ antibodies caused a graded reduction in adhesion, migration, and phagocytosis and destruction of the actin cytoskeleton. These effects were mediated by reduced Rho GTPase activity related to tyrosine 24 phosphorylation and downregulation of protein phosphatase 1B (PTP1B) activity.

### 3.4. Anti-CD40 Autoantibodies

Anti-CD40 antibodies have been studied as possible mediators of podocytopathy. Chebotareval et al. found significantly elevated levels of anti-CD40 antibodies in both MCD and FSGS patients compared to healthy controls and patients with immune complex-mediated glomerulopathies (MN, IgAN, and membranoproliferative glomerulonephritis) [85]. Interestingly, antibody levels were higher with partially responsive or resistant FSGS, as compared to steroid-sensitive FSGS or MCD.

Delville et al. obtained serum samples pre-transplantation and after 1 year from 98 adult kidney transplant recipients from four US centers, including 64 with FSGS, 33 of whom had recurrence [89]. Following microarray analysis for potential target antigens for antibody analysis, a panel of seven antibodies could predict FSGS recurrence with 92% accuracy. The best correlation was with pre-transplant anti-CD40 antibodies (78% accuracy). Both anti-CD40 antibodies purified from the sera of patients with FSGS recurrence and recurrent serum itself depolarized podocytes and reduced F-actin, an effect partially reversed by CD40-blocking antibody. Furthermore, suPAR-β_3_ integrin signaling mediated this CD40 antibody effect, as both a uPAR blocking antibody and an α_V_β_3_ blocker also reduced the anti-CD40 effect. Injection of anti-CD40 or recurrent serum into mice enhanced suPAR mediated proteinuria, an effect not seen in CD40-deficient mice or wild-type mice treated with CD40-blocking antibody.

### 3.5. Anti-Nephrin Autoantibodies

In 41 children and 21 adults with MCD from four North American institutions and the Nephrotic Syndrome Study Network (NEPTUNE) cohort, Watts et al. found 18 cases (29%) positive for anti-nephrin antibodies by indirect ELISA performed during active disease, as compared to 1 of 54 sera positive for phospholipase A2-receptor antibodies. In those with available serum during complete or partial remission, there was a complete absence of or significant reduction in circulating antibodies, respectively [8]. By immunofluorescence, punctate IgG glomerular staining that colocalized with nephrin was seen by confocal microscopy. There was no colocalization with synaptopodin or other intracellular proteins (podocin and WT1) in those with punctate staining away from the slit diaphragm. Of nine patients with punctate IgG and available serum, all nine had circulating anti-nephrin antibodies.

Fujita et al. found circulating anti-nephrin antibodies by a modified ELISA in 5/7 adult Japanese MCD patients [11]. All five had punctate IgG podocyte deposits that were colocalized with nephrin.

Using a commercial ELISA in adults, Chebotareva et al. found significantly higher anti-nephrin antibody levels in 11 patients with MCD (detectable in 10/11) and 32 with primary FSGS as compared to 25 with MN and healthy controls [10]. Antibody positivity did not correlate with response to therapy. Notably, however, 10/25 with PLA2R-positive MN had detectable anti-nephrin antibodies.

Shirai et al. studied Japanese pediatric kidney transplant recipients with FSGS and compared 11 with recurrence to 3 with non-recurrence, 8 with genetic FSGS, and 30 controls (13 healthy children and 17 with MN or lupus) [9,90]. Elevated anti-nephrin levels (threshold defined as maximum level in controls, 231 U/mL) occurred in all 11 with recurrence (831–1292 U/mL) compared to those with non-recurrence (153–244 U/mL) and genetic FSGS (33–172 U/mL). The effect of post-transplantation immunosuppression on antibody positivity is uncertain and may confound these results. Immunofluorescence microscopy on 1 hour post-transplantation biopsies of six recurrent patients revealed punctate IgG deposition that colocalized with nephrin at the slit diaphragm and intracellularly. Similar punctate colocalization was found in later biopsies during recurrence (intracellular more frequently), but in no biopsies after attaining remission. All 1 hour biopsies and biopsies during recurrence showed positive tyrosine phosphorylated nephrin and upregulated Src homology and collagen homology A staining (both mediating podocyte dysfunction), as compared to none with non-recurrence or genetic FSGS. Of 13 non-transplanted patients with MCD, 5 (38%) had elevated levels with significant reduction at remission.

Batal et al. studied 39 adult North American/European kidney transplant recipients with primary podocytopathies (7 MCD and 32 FSGS) for the ability of pre-transplant anti-nephrin antibodies to predict recurrence, which occurred in 21 patients [12]. Pre-transplant levels were predictive of recurrence (area-under-the-curve 0.78, *p* = 0.03); 8/21 with recurrence had elevated levels versus 0/17 without recurrence with a 38% sensitivity and 100% specificity. The effect of serum storage on the ability to detect these antibodies is uncertain. Staining for IgG and nephrin in 24 patients (16 recurrent and 8 non-recurrent) revealed punctate nephrin staining in 12/16 versus 1/8 (*p* = 0.008). The following four patterns emerged on dual staining: colocalized punctate IgG and punctate nephrin, non-colocalized punctate IgG and punctate nephrin, negative IgG but punctate nephrin, and negative IgG with normal linear nephrin. Non-recurrent cases had negative IgG and linear nephrin (in 7/8).

Hengel et al. found anti-nephrin antibodies in 46/105 (44%) adult Europeans with MCD, 7/74 (9%) with primary FSGS, and 94/182 (54%) of non-biopsied children with INS [13]. The prevalence was even higher in untreated adults with active nephrotic MCD (69%) and untreated nephrotic children (90%). Levels detected by a hybrid immunoprecipitation-ELISA assay correlated with disease activity over time. Immunizing mice with murine nephrin induced nephrotic syndrome, nephrin phosphorylation, slit-diaphragm IgG, and colocalization of punctate IgG-nephrin.

Using a cell-based indirect immunofluorescence method, Chen et al. found that 7/36 (19.44%) adult Chinese patients with MCD had anti-nephrin antibodies, as compared to 2/16 (12.5%) with primary FSGS and 0 of 20 with MN, 17 with diabetic nephropathy, 19 with IgAN, and 20 healthy controls; 7/26 (26.9%) actively nephrotic MCD patients were positive [14]. In the MCD group, anti-nephrin positivity was significantly associated with higher proteinuria and cholesterol (total and LDL) and lower albumin. Colocalization of IgG and nephrin was found on biopsies.

Shu et al. studied 596 adult Chinese patients with MCD and primary FSGS (436 and 160, respectively) and found anti-nephrin antibodies in 43% of the 596, including 30% with IgG, 26% with IgM, and 13.1% with both classes [15]. The prevalence was not significantly different between MCD and FSGS, and was highest in those with nephrotic syndrome not being immunosuppressed (51.1%). Positivity was associated with higher proteinuria, more relapses, and a shorter period to relapse, as compared to antibody negativity. Dual positivity was significantly more severe than having either IgG or IgM alone. In patients attaining clinical remission with available serum, antibody levels were significantly lower than while active.

Several studies have evaluated non-biopsied patients with INS. Horinouchi et al. utilized an ELISA and found anti-nephrin antibodies in 6 of 13 children during active disease that were significantly reduced following steroids [16]. In a study of 156 patients with INS, 32 transplanted patients with FSGS (+/− recurrence), and 143 controls, Angeletti et al. (published only in abstract form) did not find a significant difference between nephrosis patients and controls, nor for recurrence (versus non-recurrence) of FSGS using an ELISA (aa 1–1055) [17]. Levels did not correlate with degree of proteinuria.

Using an ELISA directed against full-length nephrin, Bianchi et al. studied 60 children with INS (20 at onset, 24 at relapse, and 16 at remission), of which 50 were steroid-sensitive. They found significantly elevated levels both at onset and at relapse versus healthy controls. Levels were significantly lower with remission and nearly undetectable in genetic cases [18].

Hengel et al. analyzed 333 nephrotic pediatric cases from an international cohort broken down by immunosuppression responsiveness, including 101 steroid-sensitive, 67 steroid-dependent, 103 non-genetic steroid-resistant, and 62 genetic steroid-resistant cases [19]. Anti-nephrin antibodies were found in 69/101 (68%) with SSNS, 19/67 (28%) with steroid dependence, 14/103 (14%) with non-genetic steroid resistance, and 1/62 (2%) with genetic steroid resistance. When sampled during active disease, 82%, 75%, 14%, and 2%, respectively, were positive, a result indicating correlation with disease activity. When assessing responsiveness to intensified immunosuppression (versus multidrug resistance) in non-genetic resistant cases, 13/74 responsive (18%) patients were antibody-positive versus 0/17 with multidrug resistance. Responsive patients had significantly lower serum albumin. The 61 anti-nephrin-antibody-negative but responsive to intensified immunosuppression patients may have had antibodies to other proteins or a non-antibody but immunologically mediated pathogenesis (*vide supra*).

Another approach has been to start with evaluation for biopsy evidence of antibody mediation. Utilizing high-resolution confocal microscopy and stimulated emission depletion (STED) microcopy, Raglianti et al. retrospectively studied 33 European pediatric patients with non-genetic steroid-resistant podocytopathy (17 FSGS and 16 MCD) and found that 16 (48.5%) had slit diaphragm IgG deposits (7/17 FSGS and 9/16 MCD), as compared to no patients with other immune-complex glomerulopathies and 1/12 with genetic podocytopathy [20]. Slit positivity was associated with the most severe nephrotic syndrome but with less chronicity on biopsy and the best long-term prognosis. Nephrin colocalized with IgG in 14/18 slit-positive cases (77.8%), a result that indicates IgG reacted with other slit-diaphragm proteins in 22.2% of cases. Similarly, in a second cohort of 40 adult patients (21 FSGS and 19 MCD), slit-positive IgG was found in 17 patients (42.5%, including 4/21 FSGS and 13/19 MCD), of which 55.6% had IgG-nephrin colocalization; 44.4% without colocalization presumably had reactivity to non-nephrin antigens. Using two ELISA assays for circulating anti-nephrin antibodies, 4/6 slit-positive cases had anti-nephrin antibodies. The two negative cases did not have IgG-nephrin colocalization. All slit-IgG-negative cases were negative in the serum. Similar results were obtained in adults. These results mirror those of Batal et al. noted above [12].

In a follow-up multi-center study, Raglianti et al. used STED microscopy to evaluate 116 kidney biopsies with primary podocytopathies and found that 44/116 (38%) had slit-diaphragm IgG deposits, including 50% of MCD and 26% of FSGS cases but 0 of 68 patients with other types of glomerulonephritis [91]. The IgG deposits colocalized with nephrin in 22/44 (50%) IgG-positive cases, with a higher incidence in pediatric patients (65%) as compared to adult cases (33%). Colocalization with podocin was found in 11/44 (25%), including 5 that also colocalized with nephrin. Kirrel1, another slit-diaphragm protein, colocalized with IgG in three patients (7%). By testing serum from a cohort of 66 patients with INS and active nephrotic syndrome for anti-podocin antibodies with an ELISA assay, 12/66 (18%) were positive, including all 7 with dual-positive IgG/podocin on biopsy. The serum was negative in 33/34 patients without biopsy IgG/podocin colocalization (97% specificity). Additionally, all 200 control patients (50 with other proteinuric and 150 with non-proteinuric kidneys diseases) tested negative. Using an anti-kirrel1 ELISA on 41 biopsy-validated cases, all 3 with colocalization of IgG/Kirrel1 were positive versus 0 without colocalization and 0 of 200 control patients.

## 4. Discussion

It has been debated for decades as to whether MCD and primary FSGS are separate entities or represent a single entity with variable severity [6]. In our opinion, neither position is correct. Neither MCD nor FSGS likely represent single pathophysiological entities. The primary process in both is podocyte injury manifested clinically as nephrotic syndrome, with or without progression towards ESKD, and pathologically as diffuse FPE, with or without segmental sclerosis. Multiple different circulating factors and/or antibodies have been proposed as pathogenic, superimposed on an at-risk genetic background. It is possible, and in fact likely, that more than one factor is involved in the individual patient. A limitation of the available studies is that the majority are single-center cohorts or retrospective analyses, which hinders generalizability and comparative analysis.

Furthermore, the same factor can produce a phenotype consistent with MCD in one patient and FSGS in another, with the former maintaining good kidney function and the latter frequently developing glomerulosclerosis, interstitial fibrosis, and loss of function. From the individual patient perspective, the most important factor is not the presence of a segment of sclerosis, but rather the clinical course, i.e., response to steroids, frequency of relapses, toxicity of steroids and other immunosuppressants, and progression to ESKD.

The evidence for a circulating factor or factors mediating primary podocytopathies is quite convincing. Injection of plasma from patients with FSGS into rats causes proteinuria [92]. Recurrence of primary FSGS following kidney transplantation may occur very rapidly, within hours, manifesting on initial biopsy as only FPE [93], and plasmapheresis can induce remission [94]. Transient proteinuria in the newborn from a mother with FSGS was reported [95]. Donation of both kidneys from a young donor that died with active MCD had complete resolution of glomerulopathy in two recipients [96]. Similarly, two recipients from a deceased donor with proteinuric FSGS had resolution of proteinuria [97]. Furthermore, resolution of nephrosis occurred upon re-transplantation of a kidney into a second recipient following refractory FSGS recurrence in the first recipient [98].

An immunologic mechanism is clearly implied by the identified antibodies and response to corticosteroids and/or other immunosuppressants. However, the main agents in use (corticosteroids [99], calcineurin inhibitors [100], and rituximab [67]) all have direct beneficial podocyte effects, independent of immunosuppression.

Underlying genetic susceptibility no doubt contributes to risk [101]. The most established genetic risk factor for FSGS is associated with apolipoprotein L1 variants (G1 and G2) found in persons of West African descent, wherein biallelic presence (G1/G1, G2/G2, and G1/G2) versus zero or one at-risk alleles markedly enhances risk by as much as 17-fold [102]. However, having two at-risk alleles is neither necessary nor sufficient to produce disease. Additional “hits” are required. Recent data from West Africa indicate that even one at-risk allele significantly increases risk for FSGS (61% higher versus zero alleles) compared to an 84% higher risk for two alleles (versus zero or one) [103]. ApoL1 functions as an ion channel, and the two at-risk variants have gain-of-function, which may explain both their enhanced trypanosomal-lytic activity and ability to damage podocytes. An oral, small-molecule inhibitor of apoL1 channel function, inaxaplin, was recently shown to inhibit channel function in vitro in human embryonic kidney 293 cells (HEK293 cells), to reduce proteinuria in transgenic G2 mice, and in a phase 2a study of 16 patients with biopsy-proven FSGS, to reduce proteinuria by 47% at 13 weeks of oral therapy [104].

As more than one factor may be involved in a given patient, potential synergy must be considered. For example, Hayek et al. evaluated the rate of decline of eGFR in two cohorts of African Americans. The rate of decline of eGFR with two at-risk apoL1 alleles, a known risk for more rapid decline in eGFR, was only significantly worse with concurrently higher suPAR levels [105]. In vitro, G1 and G2 proteins interacted with suPAR to activate α_V_β_3_ integrin more than G0, and the combination caused proteinuria in mice. Whether suPAR levels could modify response to inaxaplin in patients with at-risk alleles noted above remains to be studied. Similarly, as noted above, Delville et al. showed that either purified anti-CD40-antibodies or FSGS patient serum could disrupt the actin cytoskeleton in cultured human podocytes, an effect that could be blocked by either monoclonal uPAR-blocking antibody or an α_V_β_3_ integrin blocker. This result suggests that the suPAR-α_V_β_3_ integrin system synergizes with these pathogenic antibodies [89].

The most prominent potential target for pathogenic circulating autoantibodies is clearly the slit-diaphragm protein nephrin. Overall, positive results have been found in a significant proportion of primary podocytopathies, perhaps from one-quarter to one-half or more. Among the sub-group of actively nephrotic but un-immunosuppressed MCD patients, frequencies approach 90%. Positivity suggests more active nephrosis but a better chance for immunosuppression responsiveness and less chance for progression.

In 1999, Topham et al. demonstrated that the nephritogenic antibody mAb 5-1-6, capable of inducing severe proteinuria when injected into rats, is directed against an extracellular domain of rat nephrin [106]. Congenital nephrotic syndrome of the Finnish type, a cause of severe neonatal nephrotic syndrome, is caused by mutations in the nephrin gene (*NPHS1*) [107]. Although over 200 mutations have been found, 1 mutation (called Fin-major) leads to a severely truncated protein (90 out of 1240 amino acids) [107] and is associated with recurrence of nephrotic syndrome following transplantation. In 2002, Patrakka et al. studied 45 Finnish patients who received 51 transplants, with recurrence developing in 15/51 (29%), a rate much higher than in non-Finnish patients [108]. Recurrence only occurred with Fin-major homozygosity, with the complete absence of nephrin, and four of nine children with recurrence had anti-nephrin antibodies detected by ELISA, along with foot process effacement and no immune deposits. Immunofluorescence and immunohistochemistry staining for nephrin showed a discontinuous and coarse granular pattern, mostly intracellularly, instead of the normal linear pattern mirroring the redistribution of nephrin away from the slit diaphragm found in animal models [106], cultured podocytes exposed to nephrotic plasma [109], and biopsies of patients with MCD [110,111]. Importantly, using human anti-nephrin antibodies derived from plasma exchange fluid from a patient with active MCD, Hengel et al. induced proteinuria in a rabbit that had only FPE microscopically [112]. The totality of evidence outlined above and in the section on anti-nephrin antibodies, including their ability to produce a similar disease in experimental animals, suggests that these specific autoantibodies are indeed drivers of disease and not epiphenomena. The evidence for the other described autoantibodies is not nearly as strong and requires further confirmation.

The involvement of anti-nephrin antibodies in a significant percentage of cases has important clinical implications. The detection of these antibodies supports earlier consideration of B-cell targeted therapy such as rituximab, although this requires confirmation in randomized controlled trials. The ability to detect such antibodies must not be restricted to specialized centers, but should have widespread availability. A barrier to widespread implementation of anti-nephrin antibody screening is standardization of the ELISA assay [113,114,115]. A commercial ELISA employing the extracellular domain of human nephrin (amino acids (aa) 23–90) was used in two studies (see Table 3) [10,20]. Otherwise, in-house ELISAs were used with various targets of nephrin’s extracellular domain, including human nephrin aa 1–1059 expressed in HEK293 [8,12], human nephrin aa 23–1029 expressed in mouse cells [9,20], immunoprecipitation with human nephrin aa 36–1052 expressed in HEK293 cells followed by ELISA [13,19], human nephrin aa 23–1055 expressed in HEK293 cells [18], and indirect immunofluorescence against full-length human nephrin expressed in HEK293 cells [14,15]. Liu et al. performed comparative analysis and found the most robust results with immunoprecipitation with subsequent Western blotting, but that may be too labor-intensive for routine use [113]. A high-throughput magnetic bead-directed ELISA was found suitable for screening. Full-length nephrin may densely assemble into higher-order structures, masking potentially reactive epitopes. Furthermore, expression in mouse versus human cell lines may produce different glycosylation patterns, altering immunogenicity. To advance the field, a standardized widely available assay needs to be developed.

Another technical barrier hindering widespread evaluation for anti-podocyte autoantibodies is the unavailability of high-resolution confocal microscopy and STED microscopy. Standard immunofluorescence microscopy is likely too insensitive. As shown by Raglianti and colleagues [20,91], colocalization of punctate IgG and nephrin on the slit diaphragm and intracellularly identifies patients likely to have circulating anti-nephrin antibodies and suggests that they are more likely to be responsive to rituximab or other immunosuppression, even when steroid-resistant. Punctate IgG deposits not colocalizing with nephrin implies other antigenic targets besides nephrin, as they have demonstrated (podocin and kirrel1) [91]. Again, this suggests that rituximab should be considered sooner if antibodies are detected, regardless of target. Punctate nephrin (as opposed to linear) in the absence of IgG suggests a pathogenic non-antibody circulating factor, or possibly IgM mediation. As noted above, Batal et al. found similar results in antibody-positive cases, with only some colocalizing with nephrin [12].

A standardized method to determine the presence of a non-antibody circulating factor(s) is also required to advance the field. Various assays have been developed. The albumin permeability factor assay in isolated rat glomeruli described above is not widely available [33]. Srivastava et al. found upregulated mRNA production of pro-apoptotic genes in cultured human podocytes when exposed to plasma from recurrent FSGS post-transplantation patients and not from non-recurrent cases [116]. Den Braanker et al. developed an in vitro assay identifying increased granularity in cultured human podocytes exposed to serum from patients with recurrent FSGS after transplantation or from plasmapheresis-responsive patients with native kidney FSGS [117]. Veissi et al. utilized a series of high-throughput in vitro assays that demonstrated reactive oxygen species production, actin disorganization, and reduced podocyte viability only in the serum of immunosuppression-resistant but plasmapheresis-sensitive FSGS patients, which were not found in other nephrotic states, including MCD (active or remission), steroid-resistant nephrotic syndrome, and MN [118].

Most recently, Gupta et al. utilized a three-dimensional approach using kidney organoids (glomeruli and tubules) derived in vitro from pluripotent stem cells to determine potential circulating factors [119]. When exposed to plasma from patients with native kidney primary FSGS or recurrent after transplantation FSGS, podocytopathy, matrix deposition, fibrosis, and apoptosis developed within the organoids, a result not found with plasma from non-recurrent FSGS. Furthermore, plasma obtained after plasmapheresis treatment of recurrence led to less apoptosis. It remains to be determined whether a positive result in any of these assays could be used to predict plasmapheresis responsiveness in otherwise resistant cases.

## 5. Conclusions

It is uncertain how to integrate the plethora of data on circulating factors and autoantibodies outlined above. Ideally, a multi-center, racially admixed cohort could be screened with a battery of tests for validated circulating factors and validated autoantibodies combined with confocal and STED microscopy. Currently, only nephrin is sufficiently validated based on the multiple studies and pathogenicity in animal models. Screening for anti-nephrin antibodies seems reasonable in any patient with idiopathic nephrotic syndrome found to have MCD or primary FSGS. Regardless of the findings, steroids seem a reasonable first choice for treatment, as KDIGO recommends [120]. Evidence of anti-nephrin or other validated autoantibodies would certainly support B-cell depletion as a second choice for frequent relapses/steroid dependence, or even as a first-line therapy in those at risk for or intolerant of steroids. Observational data support the effectiveness of rituximab in native kidney podocytopathies, especially when sensitive to steroids [121,122,123,124], and for recurrence along with plasmapheresis following transplantation [5,125]. Although rituximab appears less effective in steroid-resistant cases, perhaps stratifying by direct podocyte antibody and/or circulating antibody positivity may select those more likely to respond. Circulating factor assays may indicate those more likely to respond to plasmapheresis. Randomized controlled trials targeting B-cells and/or circulating non-antibody factors are required to guide therapy, and standardized assays should be employed in such trials. The integration of genetic risk into algorithms regarding specific therapies needs exploration.

## Figures and Tables

**Table 1 antibodies-14-00082-t001:** Potential circulating factors mediating primary podocytopathies.

Vascular permeability factor Hemopexin Cardiotropin-like cytokine factor-1 Zinc fingers and homobox transcription factors/angiopoietin-like 4 suPAR Calcium/calmodulin-dependent serine protease Micro-RNAs Soluble CD40L Antibodies (see Table 2)

**Table 2 antibodies-14-00082-t002:** Autoantibody targets in primary podocytopathies.

Ubiquitin carboxyterminal hydrolase L1 Proteosome subunit alpha type 1 Annexin A_2_ CD40 Nephrin (see Table 3.) Podocin Kirrel1

**Table 3 antibodies-14-00082-t003:** Studies assessing anti-nephrin autoantibodies in primary podocytopathies.

Study/Year	Phenotype/Number	Population	Methodology	Results	Comments
Watts/2022 [8]	MCD 62	Ped/adults	IP followed by signal-enhanced in-house ELISA with human nephrin (aa 1–1059) in HEK293 cells	18/62 (29%)	Shorter relapse-free period when positive Antibodies reduced/absent with remission Punctate IgG colocalizing with nephrin by confocal microscopy and SIM
Shirai/2024 [9]	FSGS 22 All Transplanted 8 genetic, 14 presumed pFSGS FSGS 8/14 presumed pFSGS recurred	Ped	Signal-enhanced in-house ELISA with commercial nephrin (aa 23–1029) in mouse myeloma cells	All 8 recurrent had elevated levels pre-Tx or at recurrence (100%) Genetic FSGS and non-recurrent FSGS levels like controls	Antibodies decreased with remission Punctate IgG colocalizing with nephrin by confocal microscopy Increased nephrin phosphorylation with increased Src homology and collagen homology A expression
Chebotareva/2024 [10]	FSGS 41 (32 pFSGS) MCD 11 MN 25	Adults	Commercial ELISA human nephrin (aa 23–92)	22/32 (68%) pFSGS 10/11 (91%) MCD 5/9 (56%) sFSGS 10/25 (40%) MN	3 PLA2R-positive MN patients also anti-nephrin positive AUC 0.71 at cutoff 41.6 ng/mL for primary podocytopathy No difference in remission rate based on anti-nephrin
Fujita/2024 [11]	MCD 7	Adult	Modified version of ELISA Watts et al. Specific nephrin N/A	5/7 (71%)	Punctate IgG colocalized with nephrin on biopsy of all 7 positive cases
Batal/2024 [12]	FSGS 30 MCD 7 SRNS 2 All transplanted	Adult	In-house ELISA with human nephrin (aa 1–1059) in HEK293 cells	8/21 (38%) with recurrence 0/17 without recurrence	100% specificity Shorter time to recurrence if positive 12/16 (75%) recurrent had punctate nephrin staining versus 1/8 (13%) nonrecurrent IgG/nephrin colocalized in 5/16 (31%) Punctate IgG and punctate nephrin non-colocalizing in 1/16 Punctate nephrin, negative IgG in 6/16 Negative IgG normal linear nephrin in 4/16
Hengel/2024 [13]	FSGS 74 MCD 105 INS * 182	Ped/adults	IP followed by ELISA with in-house human nephrin (aa 36–1052) in HEK293 cells	46/105 (44%) MCD 7/74 (9%) FSGS 94/182 (52%) INS 1/40 non-pFSGS	90% positive if untreated active NS Positivity correlated with disease activity Immunization of mice with murine nephrin produced proteinuria, nephrin tyrosine phosphorylation, and IgG localization to slit diaphragm with no EDD
Chen/2024 [14]	FSGS 16 MCD 36 MN 20, DN 17, 19 IgAN	Adults	IIF of full-length nephrin in HEK293 cells	2/16 (12.5%) FSGS 7/36 (19%) MCD 0/56 other GN	Correlated with disease activity Colocalized IgG with nephrin in MCD patients with positive circulating antibodies
Shu/2025 [15]	FSGS 160 MCD 436	Adults	ELISA against commercial human nephrin (17757-H08H, Sino Biological, 1044 aa) Antigen inhibition ELISA IgG and IgM assayed	60/160 (37.5%) FSGS 196/436 (45%) MCD Overall, 43% positive: 30% IgG, 26% IgM, 13.1% both	51% positivity if untreated nephrosis versus 43% overall Dual IgG/IgM positivity had more severe disease Antibodies decreased during remission
Horinouchi/2024 [16]	SSNS* 13	Ped	N/A	6/13 (46%)	Levels reduced following steroids
Angeletti/2024 (Abstract only) [17]	INS 156 FSGS transplants 32 (+/− recurrence) Controls 143	Ped	ELISA against human nephrin (aa 1–1055) ELISA against human nephrin (aa 23–1029) ELISA against intracellular domain	% positive N/A	No difference in levels between INS and controls No difference in 2 extracellular domain assays No correlation with recurrence Levels decreased after treatment
Bianchi/2025 [18]	INS 60 (50 SSNS, 10 SRNS)	Ped	Indirect ELISA against human nephrin (aa 23–1055) in HEK293 cells	SSNS at onset: 65% above cutoff SRNS: 30% above cutoff	Correlated with disease activity 2/8 patients tested during relapse were positive, both became negative with subsequent remission
Hengel/2025 [19]	SSNS 101 SDNS 67 SRNS (nongenetic) 103 SRNS (genetic) 62	Ped	IP followed by ELISA with in-house human nephrin (aa 36–1052) in HEK293 cells	SSNS: 19/101 (68%) SDNS: 19/67 (28%) SRNS (non-genetic) 14/103 (14%) SRNS (genetic) 1/62 (2%)	Of non-genetic SRNS with active disease, 13/74 (18%) responders to IIS versus 0/17 non-responsive
Raglianti2024 [20]	FSGS 32 MCD 30 Other ** 71	Ped	High-resolution confocal microscopy 2 anti-nephrin assays: (1) A commercial kit against human nephrin aa 23–92 (2) Human nephrin (aa 23–1029) in mouse myeloma cells	FSGS 8/32 (25%) positive IgG at slit-diaphragm MCD: 10/30 (33%) positive Others ** 0/71 positive	In a second cohort of 48 adults slit-diaphragm IgG in 19% of FSGS and 68.4% of MCD. Nephrin colocalized with IgG in 14/18 pediatric cases but not in 4/18 (22%) compared to 44% of positive IgG cases in adults not colocalizing with nephrin

EDDs: electron-dense deposits; ELISA: enzyme-linked immunosorbent assay; FSGS: focal segmental glomerulosclerosis; HEK293: human embryonic kidney 293 cells; IIF: indirect immunofluorescence; IIS: intensified immunosuppression; INS: idiopathic nephrotic syndrome; IP: immunoprecipitation; MCD: minimal change disease; N/A: not available; NS: nephrotic syndrome; pFSGS: primary focal segmental glomerulosclerosis; SDNS: steroid-dependent nephrotic syndrome; SRNS: steroid-resistant nephrotic syndrome; SSNS: steroid-sensitive nephrotic syndrome; * non-biopsied children; ** 20 IgAN, 13 lupus/vasculitis, 14 C3 glomerulopathy, 19 others, 5 healthy children.

## Data Availability

No new data was generated to write this paper. Only existing sources were used and have been cited in the reference list.

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
