# Peer review of "Serum Factors in Primary Podocytopathies"

_2073-4468, 2025, doi:10.3390/antib14040082_

Round 1

Reviewer 1 Report

Comments and Suggestions for Authors
  • Lines 1–4: The title is clear and relevant. Consider shortening slightly to improve readability (e.g., “Serum Factors in Primary Podocytopathies”).

  • Lines 9–27 (Abstract): Well written and comprehensive. You may streamline some sentences to improve clarity and flow; for instance, “Various non-antibody mediated circulating factors have been promulgated…” could be simplified.

  • Lines 33–49 (Introduction): Good background. Consider clarifying the classification of FSGS subtypes with a brief example to aid non-specialist readers.

  • Lines 50–60: When mentioning Shalhoub’s hypothesis, include a sentence on how this theory has evolved over time to strengthen context.

  • Lines 61–64 (Tables 1–3 references): Tables are valuable. Ensure table formatting is consistent; some abbreviations (e.g., INS, pFSGS) should be defined in table legends for clarity.

  • Lines 65–156 (Non-antibody factors): This section is comprehensive. Consider condensing repetitive descriptions of methods/results to avoid redundancy.

  • Lines 157–228 (suPAR discussion): Strong coverage. Suggest clarifying contradictory findings with a concluding sentence summarizing current consensus.

  • Lines 229–238 (CASK): Very useful. Suggest citing whether this finding has been independently validated by other groups.

  • Lines 239–256 (MicroRNAs): Brief and clear. Consider adding a short note on clinical utility (e.g., potential biomarker role).

  • Lines 271–356 (Autoantibody pathogenesis, subsections 3.1–3.5): Detailed and informative. Consider shortening long sentences and using bullet points when listing study outcomes for readability.

  • Lines 460–474 (Discussion start): Excellent synthesis. Suggest explicitly stating whether authors believe MCD and FSGS should be considered one disease spectrum or distinct entities, as this remains debated.

  • Lines 515–567 (Nephrin focus): This is a highlight of the review. Suggest reorganizing subpoints to emphasize clinical implications first, followed by technical assay limitations.

  • Lines 591–608 (Conclusions): Strong conclusion. Consider adding a brief “Future directions” subsection, such as standardization of assays, clinical trials for targeted therapies, and integration with genetic testing.

  • Lines 609–616 (Author contributions/funding): Clear and standard. Ensure consistency (conflict of interest is mentioned twice).

  • References: Comprehensive and appropriate. A few references appear truncated (e.g., reference 5). Please verify completeness and formatting per journal style.

Author Response

We thanks the reviewer for helpful suggestions which improve the paper.

Lines 1–4: The title is clear and relevant. Consider shortening slightly to improve readability (e.g., “Serum Factors in Primary Podocytopathies”).

Response: change made.

Lines 9–27 (Abstract): Well written and comprehensive. You may streamline some sentences to improve clarity and flow; for instance, “Various non-antibody mediated circulating factors have been promulgated…” could be simplified.

Response:  change made. Now reads: Potential non-antibody mediated circulating factors have been identified, including cardiotrophin-like cytokine 1, soluble urokinase plasminogen activator receptor, and

Lines 33–49 (Introduction): Good background. Consider clarifying the classification of FSGS subtypes with a brief example to aid non-specialist readers.

Response: change made. Now reads: FSGS additionally has one or more segments of sclerosis/hyalinosis and has been subdivided into 5 subtypes by light microscopy, including collapsing glomerulopathy, the tip lesion, the cellular lesion, perihilar FSGS, and FSGS not otherwise specified 1.

Lines 50–60: When mentioning Shalhoub’s hypothesis, include a sentence on how this theory has evolved over time to strengthen context.

Response: change made: The search for the putative pathogenic factor(s) has since continued for MCD and has evolved to include cases with detectable sclerosis, i.e., FSGS.

Lines 61–64 (Tables 1–3 references): Tables are valuable. Ensure table formatting is consistent; some abbreviations (e.g., INS, pFSGS) should be defined in table legends for clarity.

Response: additions made.

Lines 65–156 (Non-antibody factors): This section is comprehensive. Consider condensing repetitive descriptions of methods/results to avoid redundancy.

Response: we condensed where possible

Lines 157–228 (suPAR discussion): Strong coverage. Suggest clarifying contradictory findings with a concluding sentence summarizing current consensus.

Response: following added: Overall, these conflicting data suggest that suPAR is most likely involved along with other factors (vide infra) in the pathogenesis of podocyte injury in primary FSGS. Elevated levels are not specific and are not able to differentiate primary FSGS from other glomerulopathies.

Lines 229–238 (CASK): Very useful. Suggest citing whether this finding has been independently validated by other groups.

Response: not that we are aware of.

Lines 239–256 (MicroRNAs): Brief and clear. Consider adding a short note on clinical utility (e.g., potential biomarker role).

Response: the following sentence was added: miRNAs may become useful as biomarkers for diagnosis and/or for the response to therapy, but more work is necessary.

Lines 271–356 (Autoantibody pathogenesis, subsections 3.1–3.5): Detailed and informative. Consider shortening long sentences and using bullet points when listing study outcomes for readability.

Response: we feel tables 2 and especially 3 serve that purpose.

Lines 460–474 (Discussion start): Excellent synthesis. Suggest explicitly stating whether authors believe MCD and FSGS should be considered one disease spectrum or distinct entities, as this remains debated.

Response: that section now reads: In our opinion, neither position is correct. Neither MCD nor FSGS likely represent single pathophysiological entities. The primary process in both is podocyte injury manifested clinically as nephrotic syndrome, with or without progression towards ESKD, and pathologically as diffuse FPE, with or without segmental sclerosis. Multiple different circulating factors and/or antibodies have been proposed as pathogenic, superimposed on an at-risk genetic background. It is possible, and in fact likely, that more than one factor is involved in the individual patient.

Lines 515–567 (Nephrin focus): This is a highlight of the review. Suggest reorganizing subpoints to emphasize clinical implications first, followed by technical assay limitations.

Response: the following precedes technical assay limitations: The involvement of anti-nephrin antibodies in a significant percentage of cases has important clinical implications. The detection of these antibodies supports earlier consideration of B-cell targeted therapy such as rituximab, although this requires confirmation in randomized controlled trials. The ability to detect such antibodies must not be restricted to specialized centers but should have widespread availability.

Lines 591–608 (Conclusions): Strong conclusion. Consider adding a brief “Future directions” subsection, such as standardization of assays, clinical trials for targeted therapies, and integration with genetic testing.

Response: the end of the conclusion section now reads as follows: Randomized controlled trials targeting B-cells and/or circulating non-antibody factors are required to guide therapy, and standardized assays should be employed in such trials. The integration of genetic risk into algorithms regarding specific therapies needs exploration.

Lines 609–616 (Author contributions/funding): Clear and standard. Ensure consistency (conflict of interest is mentioned twice).

Response: change made.

References: Comprehensive and appropriate. A few references appear truncated (e.g., reference 5). Please verify completeness and formatting per journal style.

Response: change made.

Reviewer 2 Report

Comments and Suggestions for Authors

Dear authors,

  1. In this review, many of the cited studies are limited to small, single-center cohorts or retrospective analyses, which reduces reproducibility and generalizability. Moreover, cohorts from Japan, Europe, and China are mixed, making it unclear whether the observed differences are attributable to genetic background or to methodological variability in the assays.

  1. In this review, the ELISA target regions and expression systems vary widely across studies (e.g., aa 1–1059, aa 23–1055, full-length, immunoprecipitation). Thus, comparing results under the broad category of “anti-nephrin antibodies” is problematic. Because the assays are not standardized, the potential for clinical application appears to be overstated. In addition, several studies do not adequately report sensitivity and specificity, leaving the robustness of the findings uncertain.

  1. The review tends to imply that the presence of anti-nephrin antibodies equates to being a disease driver. However, the possibility that these antibodies represent an epiphenomenon cannot be excluded. Furthermore, while some animal models are cited as proof of pathogenicity, antibody-induced models in mice and rabbits may not reliably reproduce the human disease or its specificity.

  1. The authors’ view that MCD and FSGS may represent a spectrum is reasonable, but the pathological diagnoses are not standardized—particularly with respect to exclusion of secondary FSGS or artifactual sclerosis. This lack of consistency may confound correlations between antibody status and clinical course.

  1. The authors note a higher prevalence of antibody positivity in recurrent cases, but potential confounders such as the effects of immunosuppressive therapy and differences in serum storage conditions are insufficiently discussed.

  1. In this review, assays with low sensitivity (e.g., 38% sensitivity, 100% specificity) are presented as predictive markers. Clinically, this is risky, as the interpretation of negative results remains problematic.

  1. The review suggests that anti-nephrin antibody positivity predicts response to rituximab; however, in the absence of prospective clinical trial data, such conclusions remain hypothetical and should be presented as such.

  1. Finally, while the review repeatedly emphasizes the need for standardized ELISAs and advanced imaging techniques such as STED microscopy, the translational strategy for clinical practice is underdeveloped. For example, the authors do not sufficiently address which patient subgroups should be screened or the cost-effectiveness of such approaches.

Author Response

We thank the reviewer for helpful comments which improved the paper

  1. In this review, many of the cited studies are limited to small, single-center cohorts or retrospective analyses, which reduces reproducibility and generalizability. Moreover, cohorts from Japan, Europe, and China are mixed, making it unclear whether the observed differences are attributable to genetic background or to methodological variability in the assays.

Response: we agree and added the following at the end of the first paragraph of Duscussion: A limitation of the available studies is that the majority are single center cohorts or retrospective analyses which hinders generalizability and comparative analysis.

  1. In this review, the ELISA target regions and expression systems vary widely across studies (e.g., aa 1–1059, aa 23–1055, full-length, immunoprecipitation). Thus, comparing results under the broad category of “anti-nephrin antibodies” is problematic. Because the assays are not standardized, the potential for clinical application appears to be overstated. In addition, several studies do not adequately report sensitivity and specificity, leaving the robustness of the findings uncertain.

Response: we agree. That is why we discuss in detail technical limitations in our discussion: A barrier to widespread implementation of anti-nephrin antibody screening is standardization of the ELISA assay113-115. A commercial ELISA employing the extracellular domain of human nephrin (amino acids (aa) 23 – 90) was used in 2 studies (see Table 3)79,90. Otherwise, in-house ELISAs were used with various targets of nephrin’s extracellular domain, including human nephrin aa 1 – 1059 expressed in HEK29377,82, human nephrin aa 23 – 1029 expressed in mouse cells81,90, immunoprecipitation with human nephrin aa 36 – 1052 expressed in HEK293 cells followed by ELISA83,89, human nephrin aa 23 – 1055 expressed in HEK293 cells88, and indirect immunofluorescence against full-length human nephrin expressed in HEK293 cells84,85. Liu et al. performed comparative analysis and found the most robust results with immunoprecipitation with subsequent western blotting, but that may be too labor intensive for routine use113. A high-throughput magnetic bead-directed ELISA was found suitable for screening. Full-length nephrin may densely assemble into higher order structures, masking potentially reactive epitopes. Furthermore, expression in mouse versus human cell lines may produce different glycosylation patterns, altering immunogenicity. To advance the field, a standardized widely available assay needs to be developed.

  1. The review tends to imply that the presence of anti-nephrin antibodies equates to being a disease driver. However, the possibility that these antibodies represent an epiphenomenon cannot be excluded. Furthermore, while some animal models are cited as proof of pathogenicity, antibody-induced models in mice and rabbits may not reliably reproduce the human disease or its specificity.

Response: we agree. The following was added: The totality of evidence outlined above in the section on anti-nephrin antibodies, including their ability to produce a similar disease in the experimental animal, suggests that they are indeed drivers of disease and not epiphenomena. The evidence for the other described autoantibodies is not nearly as strong and requires further confirmation.

  1. The authors’ view that MCD and FSGS may represent a spectrum is reasonable, but the pathological diagnoses are not standardized—particularly with respect to exclusion of secondary FSGS or artifactual sclerosis. This lack of consistency may confound correlations between antibody status and clinical course.

Response: we agree. The following was added to the Introduction: Primary FSGS is differentiated from secondary FSGS by the presence of full-blown nephrotic syndrome (hypoalbuminemia), diffuse foot process effacement (> 80%), and lack of a known secondary cause. Herein, we are considering only primary FSGS.

  1. The authors note a higher prevalence of antibody positivity in recurrent cases, but potential confounders such as the effects of immunosuppressive therapy and differences in serum storage conditions are insufficiently discussed.

Response: a valid point. The following were added at line 372 The effect of post-transplantation immunosuppression on antibody positivity is uncertain and may confound these results, and line 388: The effect of serum storage on the ability to detect these antibodies is uncertain.  

  1. In this review, assays with low sensitivity (e.g., 38% sensitivity, 100% specificity) are presented as predictive markers. Clinically, this is risky, as the interpretation of negative results remains problematic.

Response: we agree. However, the high specificity makes false positives unlikely and the test still very useful.

  1. The review suggests that anti-nephrin antibody positivity predicts response to rituximab; however, in the absence of prospective clinical trial data, such conclusions remain hypothetical and should be presented as such.

Response: we agree. The following sentence from the discussion (line 559) indicates that: The detection of these antibodies supports earlier consideration of B-cell targeted therapy such as rituximab, although this requires confirmation in randomized controlled trials.

  1. Finally, while the review repeatedly emphasizes the need for standardized ELISAs and advanced imaging techniques such as STED microscopy, the translational strategy for clinical practice is underdeveloped. For example, the authors do not sufficiently address which patient subgroups should be screened or the cost-effectiveness of such approaches.

Response: cost-effectiveness issues are beyond the scope of this paper. We feel that ant patient with idiopathic nephrotic syndrome with a pathology consistent with a primary podocytopathy should be considered for screening. The following sentence was added to the conclusion: Screening for anti-nephrin antibodies seems reasonable in any patient with idiopathic nephrotic syndrome found to have MCD or primary FSGS on biopsy.

Round 2

Reviewer 2 Report

Comments and Suggestions for Authors

Dear Authors,

I am satisfied with the revisions that have been made by the authors.